# An Online Optimal Bus Signal Priority Strategy to Equalise Headway in Real-Time

Xuehao Zhai * , Fangce Guo and Rajesh Krishnan

Urban Systems Lab, Centre for Transport Studies, Department of Civil and Environmental Engineering, Imperial College London, London SW7 2AZ, UK
* Correspondence: x.zhai20@imperial.ac.uk; Tel.: +44-(0)-7579010622

**Abstract:** Bus bunching is a severe problem that affects the service levels of public transport systems. Most of the previous studies in the field of Bus Signal Priority (BSP) and Transit Signal Priority (TSP) focus on reducing a bus delay at signalised intersections and ignore the importance of balancing the bus headways. However, since general BSP methods allocate uneven priorities for individual buses, the headways of bus sequences are prioritised or delayed randomly, increasing the likelihood of bus bunching. To address this problem and to improve the reliability of bus services, we propose an online optimisation model to determine the signal duration and splits for each traffic intersection and each signal cycle for bus priority. The proposed model is able to induce the signal timing back to a baseline when the BSP request frequency is low. Using the proposed model, a statistically significant reduction of 10.0% was achieved for bus headway deviation and 6.4% for passenger waiting times. The simulation-based evaluation results also indicate that the proposed model does not affect the efficiency of bus services and other vehicles significantly.

**Keywords:** bus signal priority; headway equalising; real-time optimisation; sumo simulation

## 1. Introduction

Bus services are one of the most convenient ways for people to travel over short and medium distances in cities. The service level of buses requires the efficiency of daily operations to keep headway regularity. Many existing studies [1–5] indicated that the primary reason for uncertainties and delays in bus services in urban areas occurs at signalised intersections. Bus Signal Priority (BSP), or Transit Signal Priority (TSP), has been widely studied in the previous literature and in practice to improve bus operation efficiency and to reduce bus delay at intersections by adjusting signal timings. This approach, however, still has the following challenges in real-world applications:

- Conventional BSP strategies generate an uneven priority effect on individual buses, which disturbs the headway of the bus fleet artificially and results in bus bunching (or platooning) [6]. The signal adjustment of a BSP might speed up an early bus whereas it might also delay a bus that is already behind schedule. To reduce the bus bunching problem, it is necessary to maintain the frequency of bus headways in a BSP scheme.
- In urban areas with high bus service regularity, signalised intersections might receive a large number of BSP requests in a short period, resulting in some signal stages that tend to maintain the state of maximum extension, leading to ineffective signal strategies.

Given this context, only a few studies have considered the problem of equilibrium of bus headways in BSP strategies [7]. To the best of the authors' knowledge, few studies have proposed online BSP optimisation strategies to address this problem in real-time scenarios and have demonstrated their schemes in a professional microscopic traffic simulation environment (e.g., VISSIM and SUMO).

Against the above background, a predictive-based BSP optimisation model is proposed to reduce the variance of bus headways without compromising the efficiency of bus services. The contributions from this paper are as follows:

- An online Mixed-integer programming (MIP) model is built to determine the signal duration and splits to balance the buses' headway for each signal cycle based on real-time traffic information, including the location of buses with priority requests, queue length and vehicle platoons for each stage.

- To offset the extension/early termination of the signal cycle, the proposed MLIP model takes appropriate elasticity into account, which induces the signal timing back to the baseline when there are few or no BSP requests. The proposed strategy can reduce the negative impact of the BSP on the signal-timing coordination of a series of intersections so that the buses can smoothly pass through consecutive intersections.

- In addition to the optimisation method to minimise bus bunching, the proposed request-based BSP model can be integrated within most traffic simulation environments for scenario evaluations. In this paper, the model is tested and calibrated in SUMO, and it is reasonable to believe that the results conducted by a well-accepted simulation software are realistic and hence producing valid results.

The rest of this paper is organised as follows: The next section summarises the existing literature in the field of bus bunching and the BSP. Section three introduces the proposed BSP model and describes the objective function and the constraints, followed by a case study, including a comparative analysis and a sensitivity analysis, in section four. Finally, section five summarises the key outcomes of this paper.

## 2. Literature Review

Maintaining a headway on a bus route is one of the key tasks for operators to keep bus systems running on schedule. The phenomenon of bus bunching, first studied by Newell and Potts [6–8], refers to two or more buses on the same route arriving at the same time at a bus stop. Among the many studies on bus bunching, most have focused on mitigating the phenomenon after its occurrence in the past several decades.

Generally, the approaches used to mitigate bus bunching and improve the reliability of bus services can be divided into two categories based on the types of strategies designed by bus operators: schedule-based methods and operational-based methods. Schedule-based methods are conventional approaches that incorporate slack time (the difference between scheduled and actual arrival time) into schedules to alleviate bus bunching and maintain regular schedules [9–12]. However, schedule-based methods are suitable for a bus system with a low density of services. For high-density services, it is difficult for buses to follow fixed timetables, especially in areas with irregular ridership and frequent traffic congestion [11]. Therefore, schedule-based methods are not effective for solving bus bunching problems practically. Operational-based methods refer to a series of control strategies, such as dynamic holding [12–14], stop-skipping [15,16] and limiting the number of people on board [17]. These strategies can overcome small perturbations and alleviate bus bunching problems, but some negative influences are generated, such as holding strategies reducing the bus speed significantly [7].

To overcome the weakness of schedule-based and operational-based methods, road network operators can control signal timings to alleviate bus bunching with bus operators [7]. Signal-timing schemes with BSP/TSP can not only improve traffic efficiency [18,19] but also improve the reliability of bus services [20,21]. For better reliability and punctuality of bus services, a series of Conditional Signal Priority (CSP) models were built to serve the bus priority requests distinctively [7,22–26]. The impacts of CSP on service reliability were investigated using a simulation platform, and the findings showed that the improvement of the CSP in bus service reliability was 3.2% when compared with a non-priority scenario [24]. However, one limitation is that bus arrival times at the following downstream intersections had not been considered in the CSP strategies. To address this limitation, multi-objective optimal CSP frameworks for maximising bus service reliability were developed where both

deviations with respect to the headway as well as corresponding additional delays induced by general traffic were considered [21–23]. Recently, a selective CSP scheme was introduced, which set bus priority only when the requests could improve service reliability [7].

Although the above studies have demonstrated the effectiveness of the existing CSP when compared to no-CSP cases, there remain several challenges in the application of the CSP in practice. Firstly, most existing strategies are often off-line calibrated using historical statistical data to evaluate the effect of signal controls [7,21–23]. Among the many examples of the CSP strategies that can respond to real-time traffic situations simulated by professional traffic simulation platforms (such as VISSIM and SUMO) have not been extensively studied. Secondly, many existing studies on the CSP focus on the improvement of buses which are behind schedule. For a bus system with high-frequency services, however, it is more practical to balance the headway when compared with keeping buses following a regular headway. Any buses that disturb the headway balance should be controlled. Finally, given that the cycle duration is a variable in these CSP studies, signal operators need to know when and how to induce the signal timing back to the baseline. Few of the above studies have discussed this practical question.

Therefore, the main objective of this paper is to propose an online optimisation model to minimise the variability of the headway of the bus sequence by adjusting signal restoration, cycle extension and splits. The specific modelling process is introduced in the following sections.

## 3. Model Formulation

### 3.1. Introduction

A simple intersection with eight phases and four stages is modelled by a precedence graph, as depicted in Figure 1. The signal plan is modelled by a dynamically allocated stage split, including the green, amber change and inter-green (all red) intervals. To balance the bus headway and to ensure restoration signal timing, we developed a Mixed-integer programming (MIP) model to determine the stage splits of each signal cycle. The main advantages include computational efficiency and optimality performance within a short period. More specifically, while maintaining the linearity of the model, we can model the process of a bus fleet passing through an intersection with consideration of the vehicle queues given the position of a bus with priority requests.

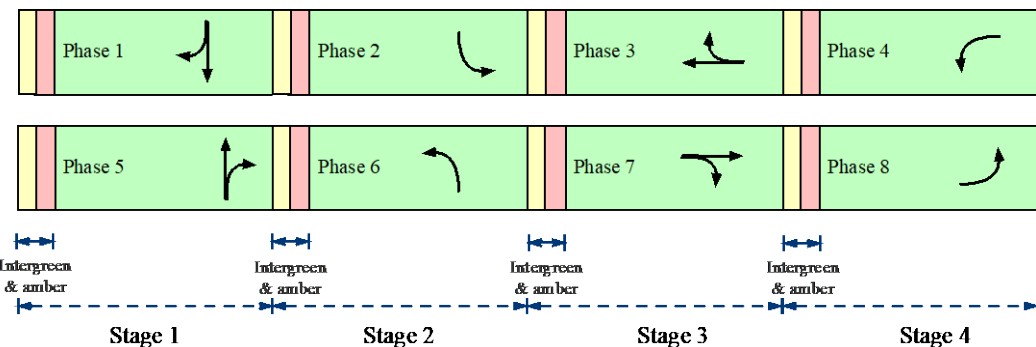

**Figure 1.** A standard stage diagram with four stages (right-hand traffic).

### 3.2. Assumptions and Notation

To facilitate describing the proposed model, we made the following assumptions:

- The location of the individual buses with BSP requests can be obtained in real-time from communication technologies, such as Vehicle-to-infrastructure (V2I).
- Buses do not overtake each other.
- All buses have sufficient capacity to carry all the passengers waiting at bus stops.
- The bus dwell time is linear and positively correlated with the number of waiting passengers at stops.

The key notations used here are summarised in Table 1, of which the dynamic parameters refer to the parameters collected or calculated at the beginning of each signal cycle, and the global parameters are set or pre-calculated beforehand.

**Table 1.** Notations and parameters.

| Sets | Subscripts | Descriptions |
|---|---|---|
| $\{C\}$ | $c$ | The set of signal cycles |
| $\{R\}$ | $r$ | The set of bus routes |
| $\{N\}$ | $n$ | The set of intersections |
| $\{S\}$ | $<r,n>$ | The set of bus stops |
| $\{P\}$ | $p,q$ | The set of Stages |
| $\{\Phi\}$ | $(c,n,r)$ | The set of BSP requests |

| Variables | Descriptions |
|---|---|
| $D_{cnr}$ | The predicted intersection bus delay of any received BSP request $(c,n,r)$ (a single bus) |
| $D^*_{cnr}$ | The difference between ideal bus delay and predicted bus delay of BSP request $(c,n,r)$. |
| $V^*_{cn}$ | The absolute between actual ending time and baseline ending time of any given signal cycle $c$ at intersection $n$. |
| $g_{cnp}$ | The green time of stage $p$ during cycle $c$ at intersection $n$ |
| $g^*_{cnp}$ | The adjustment of green signal time of stage $p$ during cycle $c$ at intersection $n$ |
| $t_{cnp}$ | The start time of stage $p$ during cycle $c$ at intersection $n$ |
| $t^0_{cn}$ | The start time of cycle $c$ at intersection $n$ |
| $\varphi_{cnr}$ | 0-1 binary variables to assign the BSP request $(c,n,r)$ (if $\varphi_{cnr} = 1$, the priority request $(c,n,r)$ is served; otherwise, the priority request $(c,n,r)$ is not served) |

| Global Parameters | Descriptions |
|---|---|
| $H'_{(r)}$ | The ideal headway on bus route $r$ |
| $g^{max}_p$ | Maximal green time of stage $p$ for each intersection $n$ |
| $g^{min}_p$ | Minimal green time of stage $p$ for each intersection $n$ |
| $L_{nr}$ | Duration of the bus $r$ travel from intersection $n$ to bus stop $<r,n>$ with an average speed |
| $\overline{g}_{np}$ | The default green time of stage $p$ at intersection $n$ |
| $M$ | A very large positive number |
| $\alpha$ | A small positive fractional number |
| $\beta$ | A positive fractional number much smaller than $\alpha$. |
| $IA_p$ | Inter-green time and amber change time between the end of stage $p$ and the start of the next stage. |
| $\theta_{cn}$ | The baseline ending time of cycle $c$ at intersection $n$. |
| $\omega_r$ | Weights for the delay variation in bus route $r$ |
| $\mu$ | Amplification parameter for bus dwelling time |
| $\varepsilon$ | Error term for bus dwelling time |

| Dynamic Parameters | Descriptions |
|---|---|
| $G^M_{cnr}$ | Needed green time to clear the moving platoon before priority vehicle with a request $(c,n,r)$ |
| $G^Q_{cnr}$ | Needed green time to clear the standing queue before priority vehicle with a request $(c,n,r)$ |
| $\overline{D}_{cnr}$ | The ideal intersection delay of the vehicle with BSP request $(c,n,r)$ |
| $A_{cnr}$ | Duration of the bus with priority request $(c,n,r)$ travel from start position (at $t^0_{cn}$) to the intersection $n$ with free-flow speed |
| $B_{cnr}$ | The departure time of the ahead bus of the one with BSP request $(c,n,r)$ at bus stop $n$ |

### 3.3. Model Formulation

In this paper, the proposed model is built to optimise the signal splits and durations. At the beginning of any signal cycle, the BSP requests, queuing lengths and signal states of the intersections are collected. All the constraints are transferred to a linear expression to make sure that a globally optimised solution can be found.

### 3.3.1. Objective Function

For any signal cycle $c$:

$$min \sum_{(c,\ n,\ r) \in \Phi} \omega_r \cdot D^*_{cnr} + \alpha \sum_{n \in N} V^*_{cn} + \beta \sum_{n \in N} \sum_{p \in P} g^*_{cnp} \tag{1}$$

The aim of the mathematical model is to minimise the total weighted variance of bus headways, the variance of the cycle ending time and the adjustment of a split. The objective function consists of three terms as follows:

The first term refers to the total variance of bus delays at the intersections. $D^*_{cnr}$ is the predicted variance of the delay of the BSP request $(c, n, r)$, which is calculated based on the difference between the predicted intersection delay $D_{cnr}$ and the ideal intersection delay, as shown in Equations (6)–(7) below. $\omega_r$ refers to the related weights of different routes. In the empirical case, all the routes are assumed equally important.

The second term refers to the sum of the variance of cycle ending times between the actual end time and the baseline, as shown in Figure 2. In the case of fewer BSP requests (assuming the first term is already satisfied), minimising the second term can induce the signal timing to restore to the baseline state, which is beneficial to keeping the offset between the upstream and downstream intersections. Hence, $\alpha$ is a relatively small number (0.5 in this study), to make sure that the restoration is conducted when the density of the BSP requests is at a lower level.

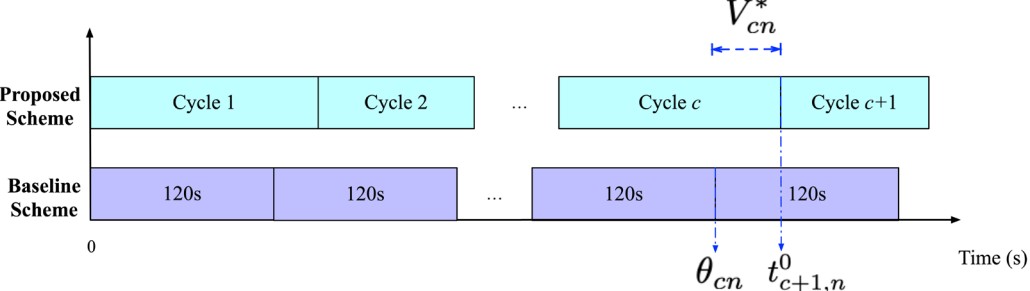

**Figure 2.** The proposed flexible signal scheme and the baseline fixed signal scheme.

The last term refers to the adjustment of the green signal. This term is added to achieve a minimum green duration change under the condition of ensuring the first two terms. $\beta$ is a generally very small positive fractional number (0.1 in this paper) when compared to $\omega_r$ and $\alpha$.

To make the objective function calculatable, the three variables $D^*_{cnr}$, $V^*_{cn}$ and $g^*_{cnp}$ of each cycle and intersection need to be expressed. $D^*_{cnr}$ is constrained based on Constraints of ideal intersection delay (Section 3.3.2) and Constraints of predicted intersection delay (Section 3.3.4). $V^*_{cn}$ is constrained based on the Constraints of restoration (Section 3.3.5). $g^*_{cnp}$ is constrained based on the Constraints of adjusting green time (Section 3.3.6).

### 3.3.2. Constraints of Stage Precedence

As the start time of the cycles and stages are varied by the influence of the signal schemes in earlier cycles, in the first step it is essential to mathematically denote the start time of any stage during a given cycle $c$, $t^0_{cn}$ and $t_{cnp}$, as followed:

$$t_{cnp} = t^0_{cn} + \sum_{q \in P(q<p)} (g_{cnq} + IA_p) \tag{2}$$

$$t^0_{c+1,n} = t^0_{c,n} + \sum_{p \in P} (g_{cnp} + IA_p) \tag{3}$$

$$t^0_{1,n} = 0 \tag{4}$$

$$g^{min}_p \leq g_{cnp} \leq g^{max}_p \tag{5}$$

Equations (2)–(5) formulate the typical four-stage signal scheme with a flexible green time as shown in Figures 1 and 2. For any intersection $n$ at cycle $c$, the green time of any stage $g_{cnp}$ is set between the default maximum and minimum values.

### 3.3.3. Constraints of Ideal Intersection Delay

As we assume that the arrival time of buses is recorded at the bus stops. For any BSP request, note that the arrival time of the ahead bus is $B_{cnr}$.

As shown in Figure 3, the bus headway (between the target bus and the ahead one) consists of the following parts: $\left(t_{cn}^0 - B_{cnr}\right) + A_{cnr} + D_{cnr} + L_{nr}$. To keep the headway close to the ideal value $H'_r$, we expected that the predicted delay $D_{cnr}$ should be close to the ideal intersection delay $\overline{D}_{cnr}$. This ideal intersection delay is expressed as in Equation (6):

$$\overline{D}_{cnr} = H\prime_k - \left(t_{cn}^0 - B_{cnr}\right) - (A_{cnr} + L_{nr}) \tag{6}$$

$$D_{cnr}^* \geq \left|\overline{D}_{cnr} - D_{cnr}\right| \tag{7}$$

$$\Rightarrow D_{cnr}^* \geq D_{cnr} - \overline{D}_{cnr} \tag{8}$$

$$D_{cnr}^* \geq \overline{D}_{cnr} - D_{cnr} \tag{9}$$

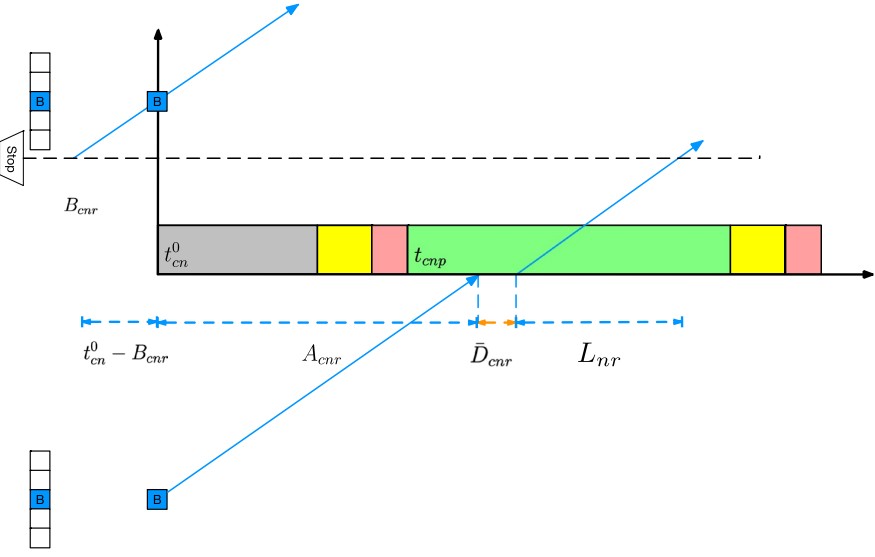

**Figure 3.** Diagram for the ideal delay evaluation.

The Equation (7) is that $D_{cnr}^*$ is equal to the absolute value of the difference between the ideal delay and the predicted delay. Equations (8) and (9) are linear expansions of Equation (7).

### 3.3.4. Constraints of Serving Priority Requests

It is necessary to make sure that the served bus with the BSP request arrives and passes the intersection in the right time window. In other words, if a BSP request $(c, n, r)$ is served, the green ending time of stage $p$ should be after the arrival time of the bus that requested it $(c, n, r)$ in free flow, which is expressed as:

$$t_{cn}^0 + A_{cnr} + D_{cnr} \leq t_{cnp} + g_{cnp} + (1 - \varphi_{cnr}) \cdot M \tag{10}$$

In Constraint (10), a binary variable $\varphi_{cnr}$ is generated to determine which request is given priority. $M$ is an exceptionally large positive number (1000 in the case study). If the $\varphi_{cnr}$ is 1, the request will only be served when the arrival time is later than the ending time of stage $p$. Otherwise, M will make the right side of the equation very large, which would result in no constraints on the variables. In other words, this constraint will be released.

### 3.3.5. Constraints of Predicted Intersection Delay

The Constraint for a predicted intersection bus delay of a BSP request $(c,n,r)$ is estimated based on Figure 4. There are three different situations of $G_{cnr}^Q$ and $G_{cnr}^M$ given a different position of the served bus as shown as Figure 4a–c.

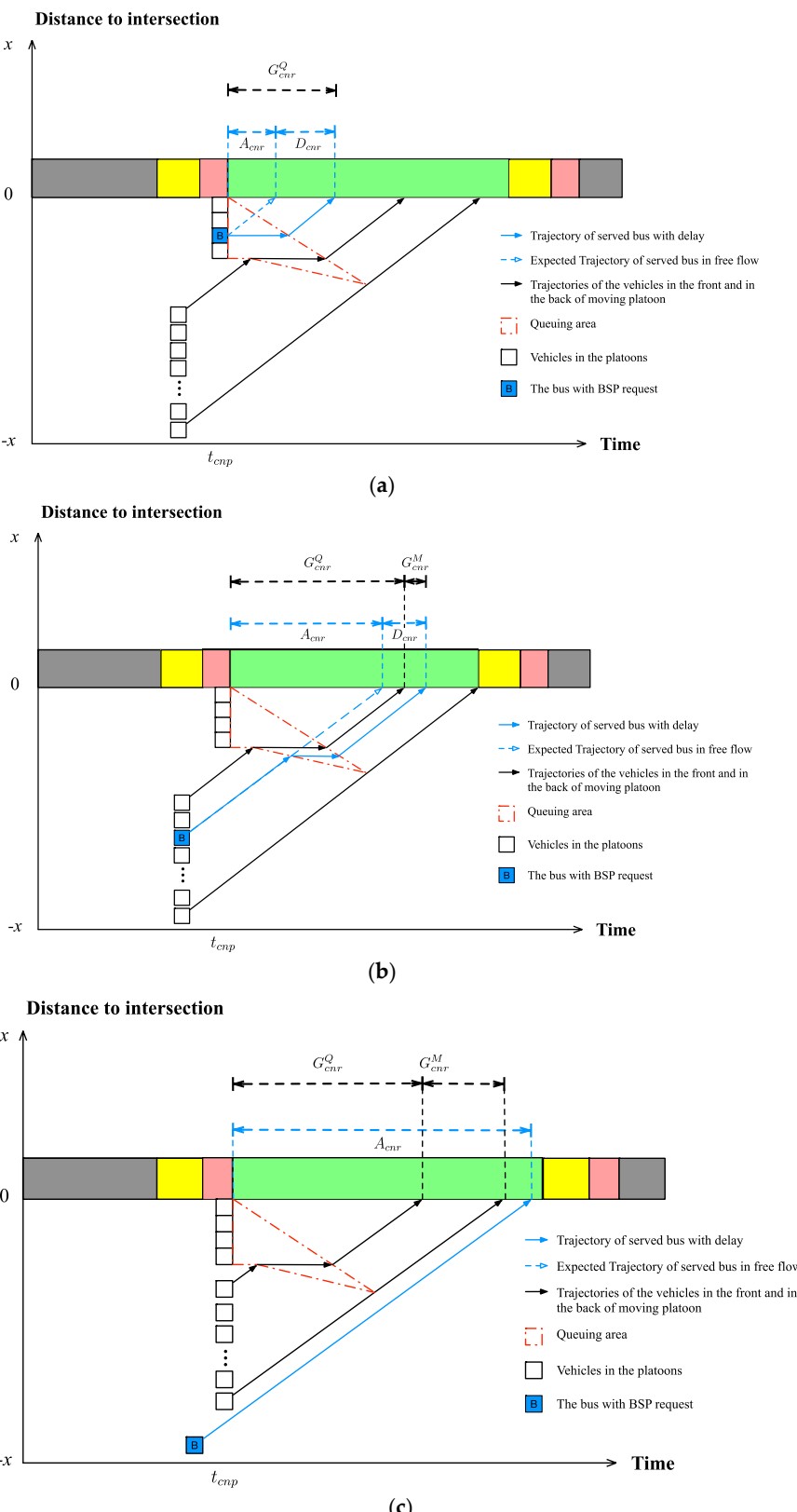

**Figure 4.** (**a**) Diagram depicting the intersection delay estimation (the served bus marked as blue block is in the middle position of standing platoon); (**b**) diagram depicting the intersection delay estimation (the served bus marked as blue block is in the middle position of moving platoon); (**c**) diagram depicting the intersection delay estimation (the served bus marked as blue block is in the end position of moving platoon).

According to Figure 4a, the solution demonstrated that $\varphi_{cnr} = 1$. As the served bus is in the standing platoon, the relevant intersection delay is only impacted by the front vehicle in the standing platoon. In this situation, the $G_{cnr}^M$ is 0. Then, $D_{cnr} \geq G_{cnr}^Q - A_{cnr}$, if $\varphi_{cnr} = 1$ and $(c, n, r)$ is in the standing platoon.

According to Figure 4b, the served bus will arrive at the intersection after the standing queue at the stop position (which needs $G_{cnr}^Q$ green time to clear) and the moving vehicles preceding the bus with a bus priority request in the platoon (which needs $G_{cnr}^M$ green time to clear). Hence, the bus delay will be larger than $G_{cnr}^Q + G_{cnr}^M - A_{cnr}$ when the priority request is served. $G_{cnr}^Q$ can be calculated by a real-time queue length estimation [27], while $G_{cnr}^M$ can be calculated by the upstream stop-bar detector and the time the priority vehicle passes the upstream intersection [27].

According to Figure 4c, the served bus will not be delayed by the platoon, which means that $G_{cnr}^Q + G_{cnr}^M - A_{cnr} < 0$. As $D_{cnr}$ is a non-negative variable constrained by Equation (16), the constraint will be released.

We can integrate the above situations to a generalized inequality constraint as follows:

$$D_{cnr} \geq \varphi_{cnr} \cdot \left( G_{cnr}^M + G_{cnr}^Q - A_{cnr} \right) + (1 - \varphi_{cnr}) \cdot (T - A_{cnr}) \tag{11}$$

$$\Rightarrow D_{cnr} \geq (-A_{cnr}) + \varphi_{cnr} \cdot \left( G_{cnr}^M + G_{cnr}^Q \right) + (1 - \varphi_{cnr}) \cdot T \tag{12}$$

Hence, Equation (11) can be simplified as Equation (12).

### 3.3.6. Constraints of Restoration

In addition to serving the BSP requests, another important ability of this method is restoration, that is, returning the signal timing back to the default state. Therefore, we estimate the bias of the ending time of the cycle and add this term to the objective function.

$$V_{cn}^* \geq \left| \theta_{cn} - t_{c+1,n}^0 \right| \tag{13}$$

Equation (13) estimates the absolute difference between the predicted ending time and the baseline ending time during cycle $c$ at intersection $n$. To induce the signal timing back to the baseline state, the second term in the objective function is designed to minimise the variance level $V_{cn}^*$.

To make the constraint in a linear expression, linearisation (same with Equations (8) and (9)) is conducted on Equation (13).

### 3.3.7. Constraints of Adjusting Green Time

The final term of the objective function is the total adjusting of green time. When there is no BSP request the signal splits should be close to the default state.

$$g_{cnp}^* \geq \left| g_{cnp} - \overline{g}_{np} \right| \tag{14}$$

Equation (14) estimates the absolute difference between the green time of cycle $c$ and the default green time. To make the Constraint into a linear expression, linear transferring, as (8) and (9), is be conducted on Equation (14).

### 3.3.8. Other Constraints

Equations (15) and (16) refer to the positive constraints of decision variables and binary Constraints of a dummy variable.

$$\varphi_{cnr} \in \{0, 1\} \tag{15}$$

$$D_{cnr}, \, g_{cnp}, \, t_{cnp}, \, t_{c,n}^0 \geq 0 \tag{16}$$

The proposed model considers either giving priority or not for each BSP request by introducing dummy variables $\varphi_{cnr}$. The advantage of this design is that it avoids the

non-linearisation associated with conditional constraints; hence, improving the computational efficiency.

## 4. Case Study

### 4.1. Simulation Scenarios

The proposed BSP model was evaluated using Simulation of Urban Mobility (SUMO) [28], an open-source microscopic traffic software package. The MIP optimisation model is programmed using the CPLEX optimiser provided by IBM. CPLEX implements optimisers based on the simplex algorithms (both primal and dual simplex) as well as primal-dual logarithmic barrier algorithms and a sifting algorithm. To interact between the real-time information (the position of the buses with priority requests and the queues in front of the them in each lane) and the online optimised scheme, the interface Traci [29,30] with Python was used to retrieve the state of vehicles and traffic lights dynamically.

A case consisting of three intersections and 20 bus stops was created to evaluate the performance of the proposed model with a BSP and a layout, which is presented in Figure 5. The eastbound-westbound road was the main road with four lanes per link; whereas, the southbound-northbound roads are branch roads with three lanes per link. General vehicles enter the simulation boundary stochastically and their arrivals follow a Poisson distribution. The maximum speed for cars and buses is 45 km/h and 35 km/h, respectively.

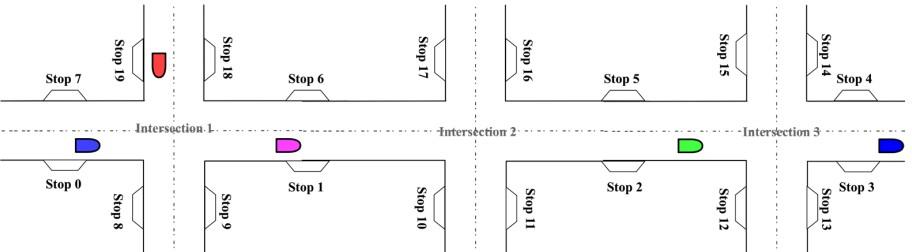

**Figure 5.** The layout of the intersections in the simulation.

With the hybrid programming of the CPLEX and Traci interfaces, the model calculation, in this case, is computationally efficient, with an optimised model calculation time of approximately 3.6 s at the start of each signal cycle.

There are 14 bus routes in the simulated case above. Operational parameters are shown in Table 2. In order to achieve a positive correlation between the bus dwelling time and the number of people waiting at the station, a series of speed control points are set after the stops. Specifically, the expression of dwelling time $DT_{cnr}$ is:

$$DT_{cnr} = DT_0 + \mu \cdot (B_{cnr} - B_{c-1,n,r}) + \tau, \ \forall c \in C, \forall r \in R \tag{17}$$

where the $DT_0$ is a constant term; $(B_{cnr} - B_{c-1,n,r})$ is the statistical time interval between two buses arriving at the station $< n, r >$. $\mu$ is an amplification parameter and $\tau$ is the error term. In this paper, $DT_0 = 10$, $\mu = 0.1$, and $\tau$ follows a normal distribution.

**Table 2.** Initial parameter settings for the network.

| Routes ID | $H'_{(r)}$ (s) | Stop Sequence | Routes ID | $H'_{(r)}$ (s) | Stop Sequence |
|---|---|---|---|---|---|
| 1 | 150 | 0→1→2→3 | 8 | 150 | 4→5→6→7 |
| 2 | 150 | 0→1→2→3 | 9 | 150 | 4→5→6→7 |
| 3 | 150 | 0→1→2→3 | 10 | 150 | 4→5→10 |
| 4 | 150 | 0→1→16 | 11 | 150 | 17→10 |
| 5 | 150 | 17→2→12 | 12 | 150 | 11→16 |
| 6 | 150 | 11→6→18 | 13 | 150 | 17→10 |
| 7 | 150 | 4→5→6→7 | 14 | 150 | 11→16 |

All bus lines are set with 150 s of departure interval. The specific routes are shown in Table 2.

For the initial signal timing, the inter-green and amber time for each stage is 3 s, the cycle time is 120 s. To evaluate the proposed model, three scenarios are introduced to conduct a comparative analysis:

- In **Scenario 1 (S1),** we apply the fixed signal scheme without the BSP.
- In **Scenario 2 (S2),** we use a standard BSP method, of which the priority strategy is red truncation. The earlier study demonstrated that the red truncation strategy has a better performance than the green stage extended strategy to reduce delays at intersections [20].
- In **Scenario 3 (S3),** we use the proposed online optimisation model with real-time information.

For each scenario, the statistical results are averaged over all five replications. To ensure the accuracy of the simulation results, the simulation results (2-4) from 400 s to 3600 s are collected to evaluate the performance of the proposed model.

Additionally, the average delay and stops times for the buses and cars, the passenger waiting time for each bus station and each bus route are selected as the measures of effectiveness (MOEs).

*4.2. Simulation-Based Evaluation and Discussion*

In this section, the variance of the headways, and the passenger waiting times of each bus route with different signal schemes is presented first. Then, the variance headways of the bus routes on the eastbound-westbound road is presented. Third, the impact of the proposed method on other vehicles is analysed based on the delay and stop times. Finally, to demonstrate the propensity of a signal cycle in the proposed method to the baseline, which is desirable, $V_{cn}^*$ as well as the BSP request frequency and any changes in the signal time at each intersection are shown.

4.2.1. The Statistical Headway and Passenger Waiting Time of Different Bus Lines

The Average Headway (AH), Standard Deviation of the Headway (STH) and (AWT) of each bus route under three scenarios are shown in Table 3. The input buses follow the Poisson distribution; both the expectation and the standard deviation of the bus headway is 150 s in this case ($\lambda = 1/150$). The passenger waiting time is shown in Table 4. Assuming that the arrival rate of passengers in any given route is $\rho$ (10 per route in a simulated minute), the passenger waiting time is calculated given the simulated bus arrival time. Note that AH is the Average Headway, STH is the standard deviation of the Headway and AWT is the Average waiting time.

**Table 3.** The average headway, the standard deviation of the headway and the average waiting time of the bus headway of 14 routes.

| | S1 | | | S2 | | | S3 | | |
|---|---|---|---|---|---|---|---|---|---|
| Routes | AH | STH | AWT | AH | STH | AWT | AH | STH | AWT |
| 1 | 202.94 | 206.56 | 91.94 | 202.81 | 215.04 | 95.17 | 203.75 | 143.99 | 87.15 |
| 2 | 317.73 | 317.73 | 121.22 | 307.36 | 346.22 | 128.08 | 328.40 | 246.08 | 107.86 |
| 3 | 149.05 | 153.05 | 70.34 | 138.64 | 181.26 | 74.08 | 150.01 | 148.10 | 67.58 |
| 4 | 190.50 | 190.11 | 87.40 | 180.22 | 212.39 | 89.08 | 196.82 | 123.13 | 88.06 |
| 5 | 127.12 | 127.08 | 128.41 | 113.12 | 153.05 | 130.11 | 128.46 | 158.53 | 93.55 |
| 6 | 126.92 | 126.89 | 62.30 | 122.89 | 166.28 | 64.90 | 127.85 | 100.63 | 62.53 |
| 7 | 152.46 | 152.59 | 67.05 | 148.82 | 168.89 | 70.00 | 151.59 | 121.93 | 68.82 |
| 8 | 156.38 | 156.38 | 96.05 | 166.38 | 186.54 | 89.97 | 157.81 | 182.56 | 84.26 |
| 9 | 147.50 | 147.50 | 66.19 | 141.50 | 162.11 | 69.23 | 144.23 | 192.61 | 67.80 |
| 10 | 257.77 | 257.15 | 134.46 | 249.36 | 283.26 | 140.64 | 260.31 | 285.55 | 120.67 |
| 11 | 138.00 | 137.30 | 68.80 | 137.35 | 130.60 | 69.55 | 138.74 | 119.12 | 69.30 |
| 12 | 125.88 | 122.29 | 58.76 | 108.29 | 157.21 | 59.22 | 122.75 | 86.22 | 59.65 |
| 13 | 116.71 | 116.46 | 53.59 | 119.46 | 114.05 | 54.28 | 117.39 | 105.30 | 53.83 |
| 14 | 120.82 | 120.56 | 58.13 | 117.56 | 111.40 | 59.40 | 122.63 | 84.84 | 58.37 |
| Total | 166.41 | 166.55 | 83.19 | 160.98 | 184.88 | 85.26 | 167.91 | 149.90 | 77.82 |

**Table 4.** The STH of buses at continuous stops on the main road.

| | The Intersections Implemented with BSP Schemes | | | | |
| --- | --- | --- | --- | --- | --- |
| Location | S1 at IS 1, 2, 3 | S2 at IS 1, 2, 3 | S3 at IS1 | S3 at IS1, 2 | S3 at IS 1, 2, 3 |
| stop 0 | 146.7 | 146.7 | 146.7 | 146.7 | 146.7 |
| stop 1 | 178.1 | 180.5 | 167.1 | 163.7 | 157.6 |
| stop 2 | 189.9 | 196.2 | 175.5 | 152.8 | 149.9 |
| stop 3 | 202.5 | 215.9 | 181.3 | 146.2 | 143.5 |
| stop 3 (exit) | 236.2 | 251.6 | 195.1 | 151.2 | 140.1 |

As shown in Table 3, several conclusions can be drawn:

- According to the results of the fixed signal scheme (S1), the sequences of each bus route keep the Poisson distribution without being influenced by the fixed signal scheme. However, in the two schemes considering BSP requests, the Poisson distribution is disrupted.
- The general BSP scheme reduces the intersection delay according to the reduction in the average headway and increases the standard deviation by 11.05% when compared with a fixed scheme. The proposed model decreases the standard deviation of the headway by 18.92% and 10.00% when compared with a general BSP and fixed scheme scenarios. In addition, the average headway is almost not affected.
- As shown in Table 4, the average waiting time with the fixed signal timing and general BSP scheme is around 83.19 s and 85.26 s, while with the proposed model it is around 77.82 s. This result complements the fact that a more uniform headway leads to a better service level and passenger service.

### 4.2.2. The Statistical Headway of Buses on the Eastbound-Westbound Road

To further analyse the effect of the proposed scheme on consecutive intersections on a mainline road, five scenarios are discussed: (a) fix scheme (S1) at all intersections; (b) General BSP (S2) at all intersections; (c) proposed scheme (S3) at intersection 1; (d) implementation of S3 at intersections 1 and 2; and (e) implementation of S3 at all intersections. The headway time distance is recorded at five locations: the entrance of stop 0, the entrance of stop 1, the entrance of stop 2, the entrance of stop 3 and the exit of stop 3.

Firstly, we can find that the headway variation in the bus fleet will naturally increase as it travels along the route (S1 fixed scheme). According to the 'S1' column of Table 4, as the bus sequences pass more intersections and bus stops, the headway will become increasingly unstable if no BSP is applied.

However, the situation will be worse when we focus only on increasing the rate of buses crossing the intersection. According to the 'S2' column of Table 4, the variation in the headway increases quickly when compared with the fixed scheme. This simulation result is consistent with our observation that general BSP schemes lead to even worse bus bunching problems.

This situation is somewhat improved when we adopt the proposed scheme (S3). When we deploy S3 in intersection 1 only (column 'S3 at IS1'), the rate of increase in STH in the bus fleet is effectively mitigated. When multiple intersections are implemented with the proposed scheme, the joint effect is significant. If the proposed BSP scheme is applied on two or more consecutive intersections, the headway standard deviation will turn towards a downward trend.

### 4.2.3. Average Delay and Stops Times

Table 5 provides the average delay and stops of the buses, car, and all vehicles systemwide in the simulation case under different control schemes.

Table 5 shows that the general BSP has a more negative impact on private vehicles, while the proposed CSP caused less car delay and simultaneously increased the efficiency of other vehicles. This is understandable because the proposed method limits the inefficient

BSP requests through the optimisation mechanism, thus improving the performance of signal timing.

**Table 5.** Comparison of the average delay and stops for vehicles and buses.

| Index | Vehicle Type | | |
|---|---|---|---|
| Intersection Delay(s) | private vehicles | buses | all |
| S1 | 42.2 | 38.9 | 39.7 |
| S2 | 44.7 (+5.9%) | 36.6 (−5.9%) | 42.4 (+6.8%) |
| S3 | 43.8 (+2.3%) | 37.3 (−4.1%) | 40.0 (+0.8%) |
| Number of Stops per intersections | private vehicles | buses | all |
| S1 | 0.98 | 0.82 | 0.94 |
| S2 | 1.01 (+5.1%) | 0.77 (−6.1%) | 0.97 (+3.4%) |
| S3 | 1.00 (+2.0%) | 0.74 (−9.2%) | 0.95 (+1.4%) |

### 4.2.4. Dynamic Restoration to Baseline (Fixed Scheme)

Over the 30 cycles of the simulation, the bias of S3 from the default signal scheme and the distribution and intensity of the BSP requests are discussed, as shown in Figure 6. In Figure 6a,c,e, the yellow bar is the bias of the cycle ending time ($V_{cn}^*$), which is calculated by the absolute difference between the actual ending time and the baseline ending time of any given signal cycle $c$. In Figure 6b,d,f, the red line (the priority of requests) refers to the sum of the absolute differences between the predicted intersection delay and the ideal delay of each BSP request in a cycle. This term is calculated as $\sum\limits_{(c,\,n,\,r)\in\Phi} D_{cnr}^*$. The blue bars (strong requests) are the frequency of the BSP requests behind schedule in a cycle ($\overline{D}_{cnr} < D_{cnr}$). The grey bars (weak requests) are the frequency of the BSP requests ahead of schedule in a cycle ($\overline{D}_{cnr} > D_{cnr}$).

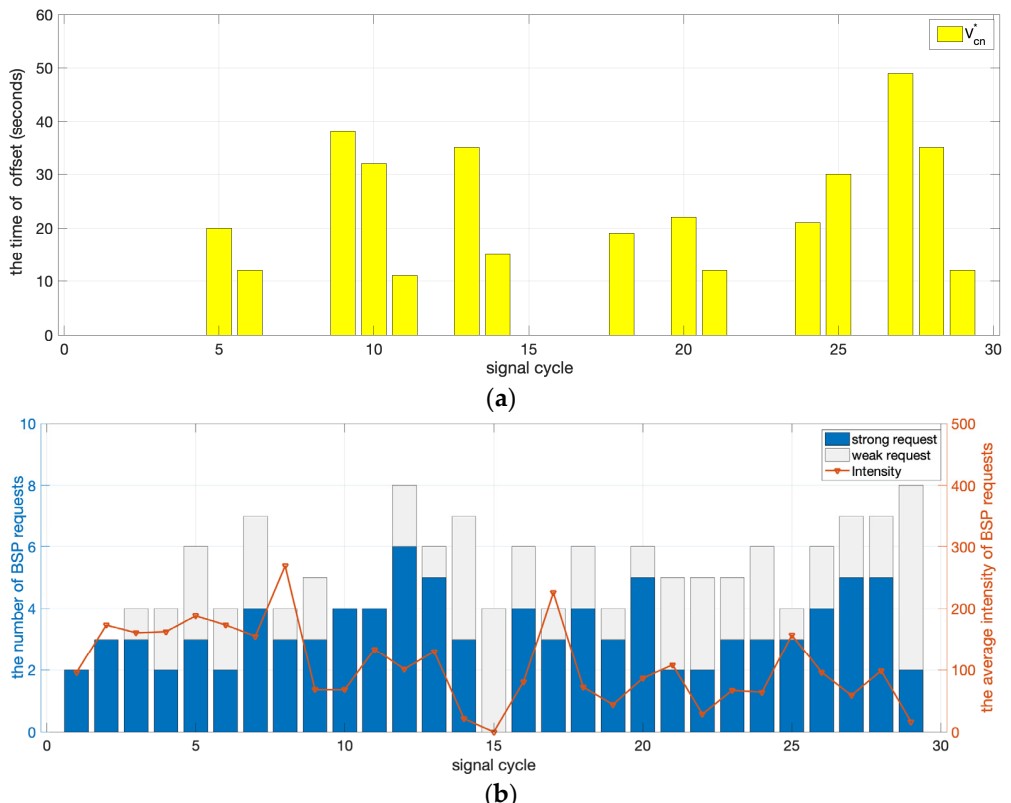

**Figure 6.** *Cont.*

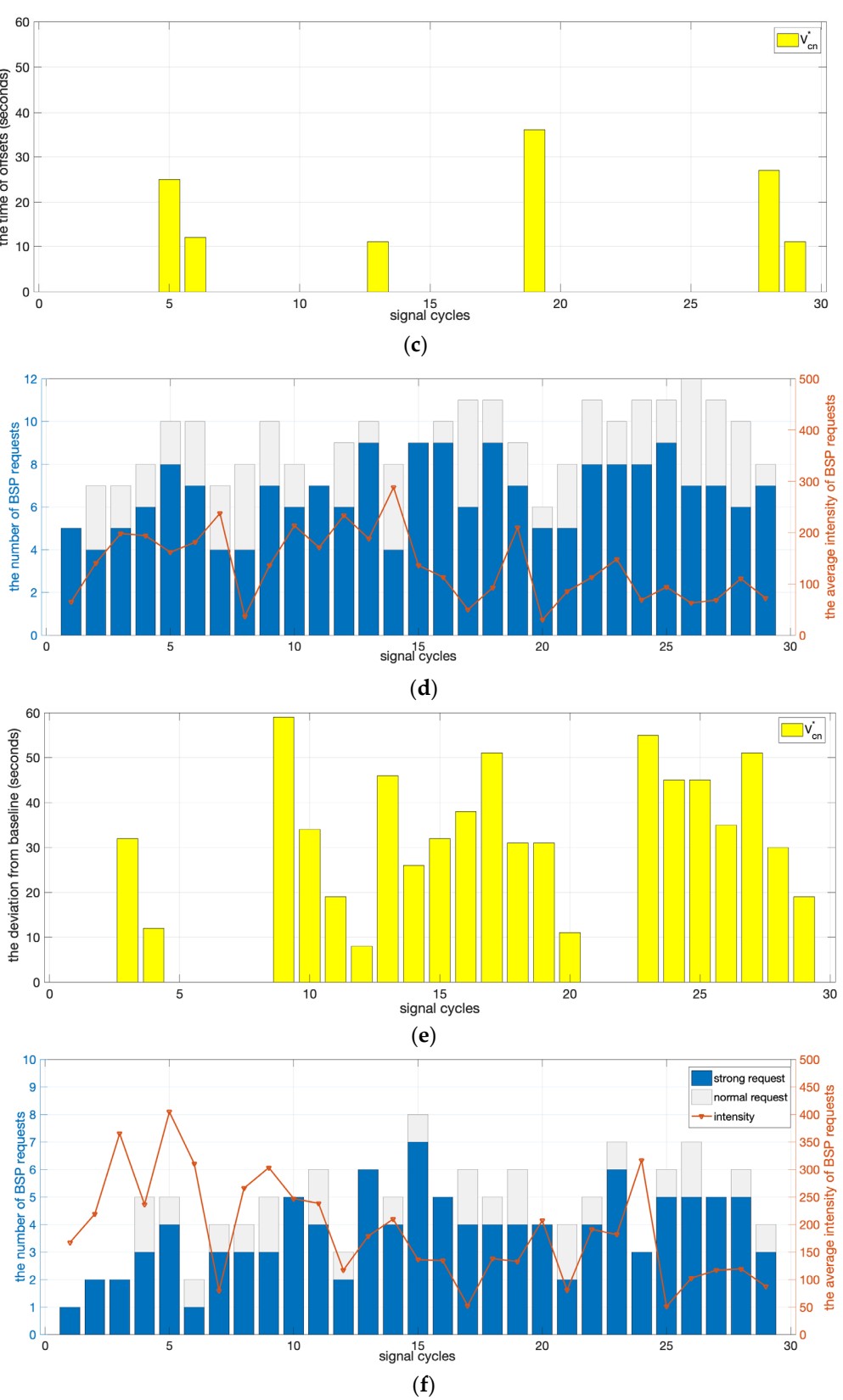

**Figure 6.** Dynamic BSP requests frequency, the priority of requests and the variance of the cycle ending time at three intersections. (**a**) $V_{cn}^*$ at intersection 1 of each signal cycle; (**b**) the BSP requests frequency and priority of requests input at intersection 1; (**c**) $V_{cn}^*$ at intersection 2 of each signal cycle; (**d**) the BSP request frequency and priority of requests at intersection 2; (**e**) $V_{cn}^*$ at intersection 3 in each signal cycle; (**f**) the BSP requests frequency and priority of requests at intersection 3.

As demonstrated in Figure 6, the resilience of the proposed model, firstly, according to Figure 6a, when $V_{cn}^*$ is relatively large, the optimised signal schemes in the following cycles will induce it to return to the baseline. Second, there is a slight correlation between the $V_{cn}^*$ and the level of BSP requests received. This is because the optimal solver can sometimes coordinate all BSP requests through a split allocation and sometimes it must solve the problem by extending the signal period. Specifically, when a small number of low-intensity BSP requests are received, the signal ending point may not return to the baseline; when many high-intensity BSP requests are received, the signal ending point may not return to the baseline. Finally, compared with intersection 1 and intersection 3, the signal scheme of intersection 2 in the middle has less $V_{cn}^*$. The reason is that the input bus sequence has a good balance, which is conducive to setting a better signal scheme (considering the balance of a bus fleet headway and the extension of the control signal period).

## 5. Sensitivity Analysis

The objective of this section is to demonstrate the performance of the proposed model under different levels of traffic demand. Four different flows are designed, and they correspond to various levels of saturation (volume/capacity ratio): 0.3, 0.6, 0.9 and 1.2. Table 6 illustrates the passenger waiting time of the proposed model under different scenarios.

**Table 6.** Sensitivity analysis results: average passenger waiting time in bus stations (s).

| | Signal Schemes | | |
|---|---|---|---|
| Saturation | S1 | S2 | S3 |
| 0.3 | 75.56 | 75.73 (+0.22%) | 73.46 (−1.50%) |
| 0.6 | 77.65 | 79.35 (+2.18%) | 74.12 (−4.55%) |
| 0.9 | 83.19 | 85.13 (+2.33%) | 77.61 (−6.71%) |
| 1.2 | 90.46 | 95.62 (+5.70%) | 82.95 (−8.30%) |

Table 6 shows that the average passenger waiting time increased gradually as the traffic demand increases. However, for all levels of traffic demand, the proposed model always outperforms both the fixed scheme and the general BSP scheme concerning the average passenger waiting times. More interestingly, the benefits increase as traffic demand increases. When the traffic demand is at the oversaturation state (i.e., the saturation rate ranges from 0.9 to 1.2), the proposed model will substantially reduce the passenger waiting time when compared to incumbent models. Wait time decreases by 6.71% to 8.30% over the fixed scheme (S1) and by 8.83% to 13.25% over the general BSP scheme (S2).

In the concluding section, the trade-off among terms in the objective functions: variance of bus headways, the bias of cycle ending times and the duration of green times are discussed. We conduct a sensitivity analysis of the amplified parameters $\alpha$ and $\beta$ (in objective function) as shown in Figure 7.

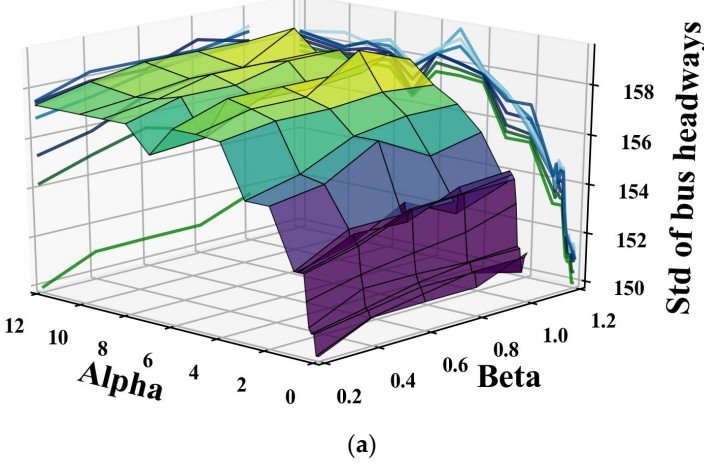

(a)

**Figure 7.** *Cont*.

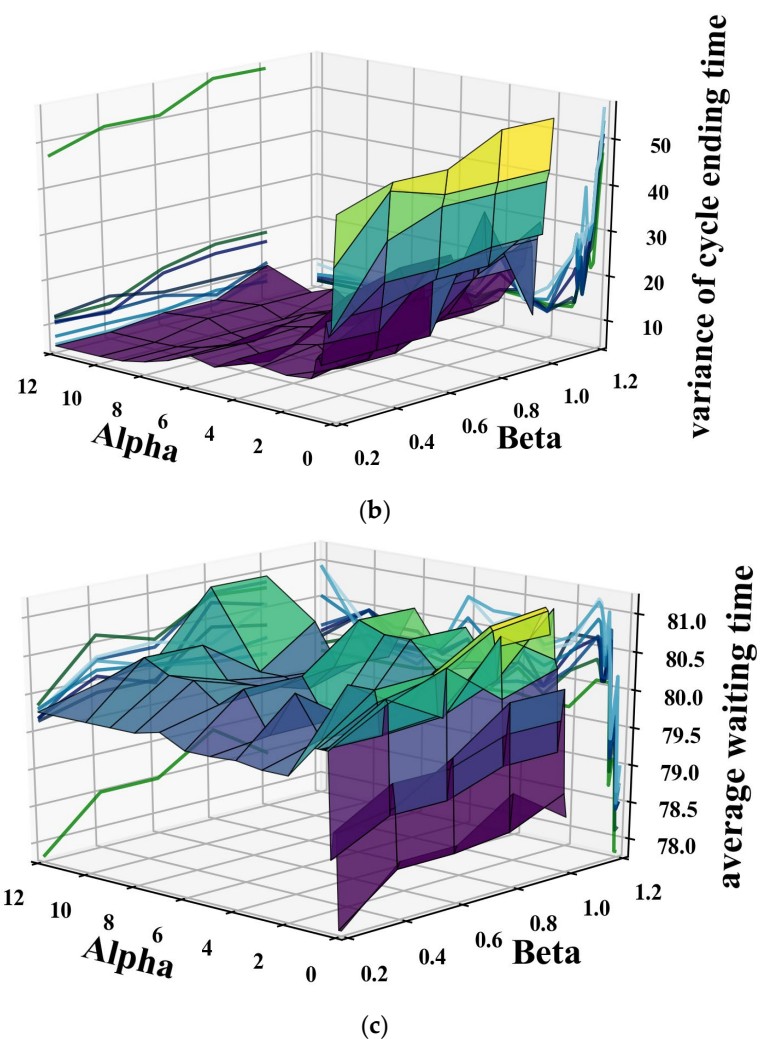

**Figure 7.** Comparison of different hyper-parameters $\alpha$ and $\beta$ on the proposed scheme S3. (**a**) STD of bus headways (STH); (**b**) bias of cycle ending times; (**c**) Average waiting times (AWT).

The parameter for signal restoration $\alpha$ is used to balance the priority on reducing the headway variance and the bias of cycle ending times. The STH and AWT increase as the amplification parameter for $\alpha$ increases. At the same time, the variance of the cycle ending time decreases. Meanwhile, the slope of the curve gradually decreases as the $\alpha$ increases. One reason for this change is that some of the BSP requests with relatively lower benefits on balancing the headway might not be met as the weight of the other terms in the objective function increases. When the $\beta$ is fixed as 0.2 and the $\alpha$ changes from a value close to 0 (0.01) to 10, the standard deviation of the headway increases by approximately 7.43 (5.1%), while the variance of the cycle ending time reduces by 41.69 s (88.38%). The result shows that if $\alpha$ takes on a significant value, S3 will prioritise keeping the signal as the default scheme. In contrast, if $\alpha$ tends to 0, S3 will continually extend the duration of the cycle when the BSP requests happen, resulting in the bias of the cycle being a substantial value (47.17). Therefore, to avoid the signal timing being overly biased towards bus services and preventing the restoration of S3, the $\alpha$ cannot take a minimal value.

When compared with $\alpha$, the effect of $\beta$ is relatively small and random. Specifically, when $\beta$ increases, AWT and STH both increase slightly. In other words, the balance of the green signal distribution of the distinct phases does not impose a significant burden on the optimal solution while the $\beta$ is relatively small (0–1).

The last finding is that the AWT varies more randomly among these indexes. The reason is that the signal scheme indirectly affects the waiting time. Factors, such as the

random distribution of the arrival of passengers and bus stopping duration may also affect the average waiting time.

## 6. Conclusions

Maintaining the headway of a bus sequence at a stable state is important for operators to maintain the reliability of bus services. This paper has developed a multi-objective CSP scheme to (1) equalise bus headways via optimising the signal splits and cycle extensions and (2) inducing the signal timing to baseline when few or no BSP requests happen. The proposed optimisation method was evaluated in a simulated area that included three intersections built in SUMO. The simulation-based evaluation results indicated that the proposed model could equalise the headways and reduce the passenger average waiting time significantly when compared with other alternatives (fixed timing scheme and BSP executing red truncation). Compared to conventional BSP schemes, the proposed scheme had less impact on other vehicles. The main reason is that this scheme filters the BSP requests and restores the default allocation in cycles when fewer BSP requests happen. With increased traffic demand, the proposed model can produce improvements in terms of reduced passenger waiting times.

The proposed BSP scheme is effective and feasible for real-world and real-time signal systems for three reasons. First, the input parameters for this scheme include the real-time location of buses and states of a vehicle platoon. This information can be practically collected from a real-world ITS system, such as automatic vehicle location data [31]. Second, the optimisation model is linear with few decision variables. Moreover, the synergy of the signal schemes between the upstream and downstream intersections is achieved using an elasticity term in the objective function. The combination of these designs ensures that the model is computationally efficient. Finally, as the signal timing is decided on an intersection-by-intersection basis, the total calculation time caused by the optimisation is linearly proportional to the number of intersections. For complex real-time simulation scenarios, the proposed scheme is still feasible.

In addition to improving service stability and reducing passenger waiting times, which are the focus of this paper, several spotlights are worthy of being targeted for BSP optimisation. Based on the combination of optimiser CPLEX and dynamic simulation using SUMO, researchers can also focus on reducing emissions [32,33] or improving the travellers' health and the environment [34]. Using the information for emissions from the SUMO simulation, the fuel and energy consumption could also be one of the attracting concerns for optimising signals for bus and private vehicles [35,36]. Last but not least, to ensure the efficiency of dynamic optimisation, the car following process and dwelling time at bus stops is simplified in this paper. This work can be improved by taking full consideration of bus travel variability [37–39].

**Author Contributions:** Conceptualization, X.Z.; methodology, X.Z.; simulation, X.Z.; formal analysis, X.Z.; writing—original draft preparation, X.Z.; writing—review and editing, X.Z., F.G. and R.K.; supervision, F.G. and R.K. All authors have read and agreed to the published version of the manuscript.

**Funding:** This research received no external funding.

**Institutional Review Board Statement:** Not applicable.

**Informed Consent Statement:** Not applicable.

**Data Availability Statement:** No new data were created or analysed in this study.

**Conflicts of Interest:** The authors declare no conflict of interest.

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
