# Peer review of "An Online Optimal Bus Signal Priority Strategy to Equalise Headway in Real-Time"

_information, doi:10.3390/info14020101_

Round 1

Reviewer 1 Report

This paper aims to propose a Mixed Integer Programming model to determine the signal duration and splits for each traffic intersection and each signal cycle for providing temporal bus priority. The model may induce the signal timing back to baseline when the priority request frequency is low. The model was evaluated by using an open-source microscopic simulator o address the optimisation problem. A case study is investigated to examine the performance of the proposed model in three scenarios: a) fixed signal scheme, b) classical priority method (ie. red truncation) and c) the proposed from the authors.

Overall, the presentation is clear, the formulation is sound, and the results support their contributions. While the paper is interesting, a few clarifications are needed:

1) Constraints of predicted intersection delay are not clearly explained. Several questions about this:

a) what if the bus does not arrive during the green? the equations should be different. in fact, there should be several different cases depending on the bus arrival time

b) not clear why there is a need to clear a platoon of moving cars if the bus is joining the back of the queue. Please explain the G_cnr variables better.

c) The lines with different colours and line types in Figure 4 are not explained, which could cause reading difficulties for readers.

2) The results are somewhat confusing - in Table 4, why does the average waiting time increase when BSP is provided in many of the cases? It's helpful to make a better clearification.

3) Similarly, in Table 3, it is not clear why the average headway increases with S3 - this again seems counterintuitive since these increase in headways actually are further than the "default" headway which I think is the goal to achieve for each route.

4) The authors usually write about bus delays. However, buses can run early-than-schedule, thus priority instead of reducing delay could be used as a holding policy to run the bus according to its schedule.

Reviewer 2 Report

In the beginning, I will write that I am a road traffic engineer with 16 years of experience. In addition to my scientific and didactic work at the University of Technology, I also design traffic lights. Therefore, some of the comments in the review are practical.

The topic taken up by the authors of the article is interesting. In my country, it has little use as buses run on timetables. Only occasional lines, e.g. on the occasion of All Souls' Day, are controlled manually by dispatchers. Then it is important to control it in such a way as to maintain constant headways between buses. However, it is worth considering using the described algorithm in places where such a solution is common.

Notes to the article:
1. Table 1 - please explain alpha, beta, and M variables in more detail. They are discussed only in the sensitivity analysis section and in this part of the article the reader does not understand what they mean. The other variables are understandable to a traffic engineer.
2. Around Figure 2 - please explain what is the impact of extending the signaling cycle on capacity. Please note that the general formula for signalized intersection capacity is C = G*g/T. As the cycle length (T) increases, the capacity of the selected approaches decreases. Please describe the compensation mechanism for disturbed cycles (if applicable).
3. Line 183 - I believe there should be a reference to figure 1.
4. Line 224 - please move the section title to the next page.
5. Line 279 & figure 1 - please specify how the time between greens is defined. In most cases, it is counted from the end of the green signal, but the picture shows that it is probably from the end of the yellow signal.
6. More data about vehicles can be obtained from the SUMO program than just the average speed in the simulation: https://sumo.dlr.de/docs/Simulation/Output/index.html If possible, I suggest obtaining data on the speeds of individual vehicles, and then compare whether they differ statistically significantly (by statistical tests). Currently, the article contains only a comparison of averages, which provides less information about the tested control method,
7. Table 5 - I suggest changing the "stop times" header to "number of stops". The current name may imply that it is the time of stops, not the number of stops.
8. Line 442 - please move the section title to the next page.
9. Line 473 - typo, redundant quotation marks at the end.
10. Please extend the bibliography and literature analysis. The article will be more valuable when current problems related to energy and fuel consumption problems are also considered. Please enter 'bus signals fuel consumption' https://www.mdpi.com/search?q=bus+signals+fuel+consumption and 'tram signals energy consumption' https://www.mdpi.com/search?q=tram+signals+energy+consumption in the search engine at mdpi.com. There are few articles on the priority of public transport at traffic lights. It is worth pointing out as a direction for further research, especially since SUMO currently allows for the analysis of emissions.
11. As a limitation, it is worth pointing out that you assumed to know the position of the vehicle at all times. In fact, in the systems operating in my country, this item is usually transmitted with a delay of a few or more seconds. It is also worth taking into account in future research erroneous GPS positions that appear in the systems used.

Good luck! I hope to read the revised article soon!
